# Recent Advances with Precision Medicine Treatment for Breast Cancer including Triple-Negative Sub-Type

**DOI:** 10.3390/cancers15082204

**Published:** 2023-04-08

**Authors:** Md Abdus Subhan, Farzana Parveen, Hassan Shah, Satya Siva Kishan Yalamarty, Janaína Artem Ataide, Valdimir P. Torchilin

**Affiliations:** 1Department of Chemistry, Shahjalal University of Science and Technology, Sylhet 3114, Bangladesh; 2Department of Pharmaceutics, Faculty of Pharmacy, The Islamia University of Bahawalpur, Bahawalpur 63100, Pakistan; 3Department of Pharmacy Services, DHQ Hospital Jhang 35200, Primary and Secondary Healthcare Department, Government of Punjab, Lahore 54000, Pakistan; 4CPBN, Department of Pharmaceutical Sciences, Northeastern University, Boston, MA 02115, USA; 5Faculty of Pharmaceutical Sciences, University of Campinas, Campinas 13083-871, SP, Brazil; 6Department of Chemical Engineering, Northeastern University, Boston, MA 02115, USA

**Keywords:** breast cancer, precision medicine, targeted therapy, metastatic breast cancer, triple-negative breast cancer

## Abstract

**Simple Summary:**

The progress of next-generation sequencing technologies has raised huge expectations for precision-medicine therapy approaches in breast cancer and triple-negative breast cancer. Targeted therapy strategies such as ICIs, EGFRi, PARPi, ADCs, CD44i, OVs, and GLUT1i are innovative therapy options for BC and TNBC. Targeting signaling pathways may also represent a prospective approach for breast cancer therapy. Combination therapy strategies can potentially enhance the precision-medicine treatment of metastatic breast cancer and TNBC patients.

**Abstract:**

Breast cancer is a heterogeneous disease with different molecular subtypes. Breast cancer is the second leading cause of mortality in woman due to rapid metastasis and disease recurrence. Precision medicine remains an essential source to lower the off-target toxicities of chemotherapeutic agents and maximize the patient benefits. This is a crucial approach for a more effective treatment and prevention of disease. Precision-medicine methods are based on the selection of suitable biomarkers to envision the effectiveness of targeted therapy in a specific group of patients. Several druggable mutations have been identified in breast cancer patients. Current improvements in omics technologies have focused on more precise strategies for precision therapy. The development of next-generation sequencing technologies has raised hopes for precision-medicine treatment strategies in breast cancer (BC) and triple-negative breast cancer (TNBC). Targeted therapies utilizing immune checkpoint inhibitors (ICIs), epidermal growth factor receptor inhibitor (EGFRi), poly(ADP-ribose) polymerase inhibitor (PARPi), antibody–drug conjugates (ADCs), oncolytic viruses (OVs), glucose transporter-1 inhibitor (GLUT1i), and targeting signaling pathways are potential treatment approaches for BC and TNBC. This review emphasizes the recent progress made with the precision-medicine therapy of metastatic breast cancer and TNBC.

## 1. Introduction

Precision medicine is an approach to the treatment of disease while considering inconsistencies in the genes, environment, and lifestyle of each person. Recent developments in omics technologies have focused on a more precise approach for precision therapy [1,2]. The precision-medicine approach is based on the selection of suitable biomarkers to predict the efficiency of targeted therapy in a specific group of patients [3,4]. In breast cancer, specific druggable mutations have been identified [1]. ESCAT (ESMO scale for clinical actionability of molecular targets) is a guide for the choice of treatment and execution of precision medicine for clinical cancer therapy. ESCAT ranks a molecular target in six different classes on the basis of the clinical indication of actionability in a specific cancer type and assists clinicians in making therapeutic decisions. For breast cancer, the most relevant molecular targets are ranked according to ESCAT as shown in Figure 1. Erythroblastic oncogene B (ERBB2), phosphatidylinositol-4,5-bisphosphate 3-kinase, catalytic subunit alpha (PIK3CA), and germline BRCA1/2 alterations are biomarkers confirmed in breast cancer, prospective for the choice of targeted therapies, which are categorized as the highest level of evidence: tier Ia. Agnostic biomarkers such as microsatellite instability and NYRK fusion show similar functions across different tumor types, which are ordered as tier Ic. In tier I, genomic alterations (GAs) are clinically relevant and should be applied in clinical use, since a clinical study has established a statistically substantial and clinically relevant survival benefit of a certain GA-TT (targeted therapy) combination. The difference between tiers Ia, Ib, and Ic is the level of evidence, such as the study design of the clinical trial in which the biomarker was analyzed (Ia: prospective, randomized; Ib: prospective, single arm; and Ic: basket trial) [5].

The molecular changes that still require further studies for confirmation are ranked as tier II. Tier-III-graded molecular changes are those for which there is proof of actionability in other tumors (IIIa) or those which have similar functional effects in the same gene or pathway of tier I with no available clinical data (IIIb). Tier IV are grouped molecular alterations for which only preclinical studies exist [3].

Breast cancer is classified into three key subtypes based on hormone receptor (HR) and human epidermal growth factor 2 (HER2) status: HR+/HER2−, HER2+, and triple-negative breast cancer (TNBC). Therapeutic selection in breast cancer has been directed mainly by this subtyping without considering the heterogeneity of the molecular scenery [6]. Six intrinsic subtypes of breast cancer include luminal A, luminal B, HER2-enriched, basal-like, claudin-low, and normal-like [7].

Endocrine therapy of breast cancer represents a personalized and continuously improving therapeutic strategy of HR+ breast cancer since the discovery of HER2. HER2 discovery has permitted the advancement of drugs that significantly enhance the outcome of therapy in HER2+ breast cancer patients [8,9,10,11]. Classic histologic strategy, multigene expression assays, and immune profiling have crucial roles in breast cancer treatment. However, significant interest has grown from the recognition and evaluation of the clinical importance of novel molecular modifications that could be a prospective therapeutic target [3].

In luminal A breast cancer, *PIK3CA* mutations are present in 45% of tumors, and *MAP3K1, GATA3, TP53,* and *CDH1* alterations are also frequently observed. In the luminal B subtype, *TP53* and *PIK3CA* mutations are present in 29% of patients. In the HER2+ subtype, ERBB2 (80%), TP53 (72%), and PIK3CA (39%) mutations are frequently present. In basal-like tumors, TP53 mutations are present in 80% of tumors, and a low occurrence of PIK3CA (9%) mutations is observed [12]. The establishment of crucial gene expression assays associated with disease recurrence may be prospective to start treatment precisely and tailor the adjuvant therapy to reduce treatment failure in breast cancer therapy. Oncotype DX, MammaPrint, Prosigna, the breast cancer index, and EndoPredict are the frequently utilized multigene expression assays for breast cancer treatment [3].

TNBC is highly diverse disease with various histopathological landscapes, driven by discrete molecular alterations. An attitude towards adapting personalized and efficient therapies for each patient is crucial for TNBC due to the increased risk of disease recurrence and mortality. The frequently utilized treatment option for TNBC is chemotherapy, which is usually associated with off-target toxicity and drug resistance. Neoadjuvant chemotherapy is a better option as it offers close monitoring of early therapy responses and affords significant prognostic evidence. Patients who have a complete pathological response (pCR) after neoadjuvant chemotherapy may have a substantially enhanced outcome. Contrariwise, poor respondents may face a higher risk of disease relapse and death. It is thus crucial to recognize the subgroups that are more likely to benefit from a personalized strategy of the available targeted therapies [13].

Platinum (PT)-based chemotherapies, DNA damage response inhibitors (DDRis), ICIs, inhibitors of PI3K/AKT/mTOR, and androgen receptor (AR) pathways are some of the progressively utilized therapies for TNBC patients.

The development of next-generation sequencing technologies has raised hopes for precision-medicine treatment strategies in TNBC. Clinical trials planned to treat TNBC patients based on subtype-specific classifications have demonstrated potential [14]. However, tumor heterogeneity and sub-clonal evolution in primary and metastatic TNBC are challenging to plans for adaptive precision-medicine-based treatment approaches.

Precision medicine, particularly genome medicine in breast cancer, is an important approach in breast cancer therapy. Several multiplex gene panels are available for targeted therapy [1]. Mutational variations and molecular targeted agents are crucial in precision-medicine breast cancer therapy development.

Advances in molecular medicine have been crucial in precision medicine. HER2 and EGFR are overexpressed in breast cancer. Trastuzumab mAbs are effective against HER2+ breast cancer. Trastuzumab binds to the extracellular domain of HER2. The combination of trastuzumab with taxanes, platinum compounds, and vinorelbine is safe and has demonstrated promising efficiency. The combination of taxane with trastuzumab is the best first-line therapy option for metastatic breast cancer overexpressing HER2. The FDA has approved trastuzumab in combination with chemotherapy for HER2+ breast cancer as an adjuvant therapy. Lapatinib is a tyrosine-kinase inhibitor (TKI), orally administered, which is effective against both HER2- and EGFR-overexpressing breast tumors. Lapatinib binds reversibly to the ATP-binding site of both receptors and inhibits receptor phosphorylation and activation. Lapatinib impedes proliferation in breast cancer and inhibits the activation of EGFR, HER2, AKT, and ERK1/2 both in vitro and in vivo [1,15].

PI3K inhibitors such as buparlisib are in the advanced stages of therapeutic development for breast cancer therapy. During a phase II study comparing buparlisib with a placebo in combination with fulvestrant in postmenopausal HR+ and HER2− breast cancer advanced during or after aromatase inhibitor therapy, buparlisib improved the median PFS (progression free survival) from 5 to 6.9 months [16]. In a similar phase III study with HR+ and HER2− breast cancer patients with disease which reverted during or after an mTOR inhibitor, buparlisib increased PFS from 4 to 6.8 months [17].

The mTOR inhibitor everolimus has been approved and frequently utilized in combination with the aromatase inhibitor exemestane in patients with HR+ progressed breast cancer after nonsteroidal aromatase inhibitor therapy [18]. PFS was enhanced from 2.8 to 6.9 months. For the exploration of the prognostic factors of everolimus, next-generation sequencing was executed to examine genetic changes utilizing archived tumor samples from a previous study [1,19]. The advantage in PFS with everolimus was reliable across the gene modifications in epidermal growth factor receptor 1 (EGFR1), cyclin D1 (CCDN1), and PIK3CA or relevant pathways.

Many AKT inhibitors have been studied in breast cancer patients. In a phase II study, ipatasertib was assessed in combination with paclitaxel for locally advanced or metastatic TNBC patients [20]. The median PFS was increased from 4.9 to 6.2 months compared to a placebo. In another study, the utilization of the AKT inhibitor AZD5363 demonstrated a PFS of 5.5 months in patients with AKT1E17K-mutant ER+ breast cancer [21].

p38 MAPK plays a crucial role in regulating cytokines, stress, and cell survival. Ralimetinib is an inhibitor of p38 MAPK, which has been tested for breast cancer patients [22]. A phase II clinical trial of tamoxifen and ralimetinib in progressed or metastatic breast cancer after aromatase inhibitors is under clinical trial [23].

Entinostat is an oral histone deacetylase inhibitor (HDACi) for the treatment of endocrine therapy-resistant ER+ breast tumors. During a phase II study investigating entinostat in combination with exemestane in contrast with exemestane only, entinostat enhanced PFS from 2.3 to 4.3 months [24].

The cyclin D-CDK4/6 inhibitors prompt cell cycle arrest in G1 phase and avert cancer cell proliferation [25]. The efficiencies of CDK4/6 inhibitors such as palbocicilib, ribocicilib, and abemacicilib in patients with ER+ breast cancer is being assessed in clinical trials [26,27,28,29,30].

Germline BRCA mutations, gBRCA, are observed in 5 to 10% of breast cancers [31]. PARP inhibitors induce synthetic lethality in BRCA1/2 defective cells. Synthetic lethality is a situation in which mutations in two genes together cause cell death, whereas mutation in either gene alone does not. Cancer cells that have only one mutated gene in a specific pair of genes may depend on the normal partner gene for existence. Synthetic lethality occurs where the individual loss of either gene is compatible with life; however, the simultaneous loss of both genes causes cell death. The genetic interaction between PARP and BRCA can be described as synthetic lethal. PARP inhibitors result in an increase in single-strand breaks (SSBs), which are transformed to irreparable toxic double-strand breaks (DSBs) during replication in BRCA1/2 defective cells [32].

The PARPi olaparib was studied in a phase III clinical trial as a monotherapy, in comparison with chemotherapeutics (vinorelbine, eribulin, capecitabine) in a patient with g-BRCA-mutated HER2− metastatic breast cancer who had received no more than two lines of chemotherapy. The PFS of the patient was increased from 4.2 to 7.0 months. Talazoparib was assessed in a phase III trial in patients with advanced breast cancer and gBRCA mutation [33]. Talazoparib is a dual-effect PARPi demonstrating inhibition of PARP enzymes and also trapping them in the DNA, impeding DNA damage repair, leading to an inhibition of BRCA-mutated cells. PARP inhibitors trap the PARP1 and PARP2 enzymes at sites of damaged DNA. The trapped PARP–DNA complexes were found to be more cytotoxic than unrepaired SSBs caused by PARP inhibition; PARP inhibitors may act as poisons that trap PARP enzymes on DNA [34]. Talazoparib improved PFS from 5.6 to 8.6 months compared to a placebo. Further, to test BRCA mutations, BRCA analysis CDx was approved by the FDA as a companion diagnostic [35]. This study discusses the recent advances and challenges with potential precision-medicine therapy strategies for metastatic breast cancer and TNBC.

## 2. Progress and Challenges of Precision-Medicine Therapy of Breast Cancer

Breast cancer is a heterogeneous disease and has different molecular subtypes with distinct clinical implications. The first molecular subtype classification of breast cancer was proposed by Perou et al. in 2000 with six intrinsic subtypes, which benefited effective breast cancer therapy development strategies [7]. Despite undesirable results, partly due to imprecise prevailing views and technology confines, precision medicine remains an essential source to lower the off-site toxicities of chemotherapeutic agents and maximize the patient benefits. The precision-medicine strategy can identify effective biomarkers which predict the efficiency of targeted therapy in a particular patient population [4]. During the past decade, the application of targeted therapy has received a lot of keen interest from researchers in the field of oncology. Among breast cancer therapies, endocrine therapy targets the estrogen and progesterone receptors for hormone-receptor-positive breast cancer. Both of the mentioned therapies have revolutionized the treatment outcomes for breast cancer patients and are widely used to target hormone receptors that are found on breast cancer cells. With the progress and advancements in precision medicines for breast cancer, the discovery of HER2 overexpression has led to the development of multiple HER2-targeted agents [36].

### 2.1. Oral HER2-Targeting Tyrosine Kinase Inhibitors

HER2 is a suitable target for inhibitory drugs such as TKI which are commonly used in a wide range of tumors. During 2020, lapatinib was the only FDA-approved HER2-specific TKI; since then, neratinib has received FDA approval, in addition to others including pyrotinib, tucatinib, and afatinib. Neratinib was primarily established as an irreversible HER1-, HER2-, and HER4-blocking TKI. It has substantial single-agent activity in HER2-positive breast cancer but has been associated with a relatively high dose of diarrhea. The toxic effect may be mitigated by prophylactic use of antidiarrheal treatment. Moreover, neratinib was recently evaluated in phase III trials in patients with early-stage HER-positive breast cancer who had already completed the trastuzumab-based standard therapy [37,38]. Pyrotinib is small, irreversible inhibitor of HER1, HER2, and HER4 molecules that has shown a good preclinical activity in inhibiting HER2-mediated downstream signaling and tumor growth in breast cancer cell lines and xenograft models [39]. A randomized phase III study compared capecitabine versus either lapatinib or pyrotinib in patients with HER2-positive breast cancer or metastatic breast cancer who previously received trastuzumab, taxanes, and/or anthracyclines. In the study, 267 patients were randomized to receive either pyrotinib or lapatinib with capecitabine. The treatment of pyrotinib plus capecitabine was associated with a higher progression free survival compared to lapatinib and capecitabine therapy. The grade of diarrhea reported was 30.6% and 8.30% [40].

### 2.2. HER2 Monoclonal Antibodies

HER2, a member of the HER family, is overexpressed in 20% of breast cancer patients. Prior to the availability of HER2-directed monoclonal antibodies, the prognosis of HER2-breast cancer was very poor. Several novel HER2-targeting monoclonal antibodies have been designed that bind to the HER2 receptor with a greater specificity than trastuzumab, or that have the ability to bind to additional epitopes to improve activity or to elicit a greater immunologic response [41]. Trastuzumab was the first monoclonal antibody for metastatic breast cancer and early breast cancer, and it has improved survival outcomes in various randomized prospective clinical trials. Margetuximab is the chimeric monoclonal anti-HER2 derivative of trastuzumab with a constant Fc region. It was engineered for mediating antibody-dependent cellular cytotoxicity via CD 16a and showed a relatively low—albeit noteworthy—advancement in PFS in pre-treated HER2-positive metastatic breast cancer in the SOPHIA trial. The SOPHIA study was a randomized phase III study that evaluated margetuximab and chemotherapy compared with trastuzumab and chemotherapy in 530 patients with HER2-positive advanced breast cancer or metastatic breast cancer who had previously received two lines of anti-HER2 therapy [42]. ZW25 is an anti-HER2 bispecific antibody targeting two HER2 epitopes (ECD2 and ECD4). It is also effective in breast cancer xenografts with low HER2 and has improved the median overall survival rate. It is under phase I and II trials in patients with advanced HER2-expressing cancer [43,44]. PRS-343 is another bispecific anti-HER2 monoclonal antibody targeting HER2 and CD137. CD137 is a potent costimulatory immunoreceptor and member of the tumor necrosis factor (TNF) receptor family. PRS-343 helps the clustering of CD137 by bridging CD137-positive T cells to HER2-positive cancer cells, leading to the increased stimulation of tumor antigen-specific T cells. The drug is currently being evaluated as a single agent and also in combination with atezolizumab in phase I trials, and both trials have allowed for patients with advanced solid tumors [41].

### 2.3. Antibody-Drug Conjugates

Antibody–drug conjugates (ADCs) are immunoconjugates comprised of a monoclonal antibody tethered to a cytotoxic drug via a linker. ADCs are designed to selectively deliver the payload directly to the target site [45]. The treatment of HER2-positive breast cancer is being revolutionized by ADCs; they are efficiently used to deliver the cytotoxic agent by using monoclonal antibodies with reduced off-side toxicities, and examples include trastuzumab deruxtecan, ARX788, and ZW49. Trastuzumab deruxtecan is a novel HER2-targeted ADC comprised of trastuzumab, a cleavable drug linker, and topoisomerase I payload (exatecan derivative) and has received accelerated approval by the FDA for the treatment of patients with HER2-positive, unresectable breast cancer who have received at least two prior lines of anti-HER2-based treatment protocols [46]. ZW49 is another antibody drug conjugate comprised of ZW25 combined with monomethyl auristatin E. A number of in vitro studies have demonstrated that ZW49 is internalized rapidly by HER2-expressing cells compared with monospecific trastuzumab-ADCs. The in vivo antitumor activity of ZW49 in patient-derived xenograft models has been demonstrated in cell lines with high and low HER2 levels [47]. On 5 August 2022, the ADC drug fam-trastuzumab deruxtecan-nxki (Enhertu) received FDA approval for unresectable or metastatic HER2-low breast cancer patients who have received prior chemotherapy for metastatic disease or who developed disease recurrence during or within six months of completing adjuvant therapy [48]. On 3 February 2023, the US-FDA approved the ADC drug sacituzumab govitecan-hziy (Trodelvy, Gilead Sciences, Inc.) for the therapy of patients with unresectable locally advanced or metastatic hormone-receptor-positive (HR+), and HER2- metastatic breast cancer patients who received endocrine-based therapy and at least two more systemic therapies for metastatic disease. Therefore, ADC drugs are promising for the treatment of HER2-metastatic breast cancer patients.

### 2.4. Challenges in Precision Medicine

Since the FDA approval of trastuzumab in 1998, the treatment options for patients with HER2-positive breast cancer have undergone a significant shift in the field of precision medicines. Various drugs have been approved by the FDA, and there are other prospective new drugs in the pipeline that have exhibited a good clinical function in HER2-positive breast cancer. Given the quickly growing therapeutic arsenal, the potential strides over the next several years should reveal the optimal sequencing of the various treatment options that have maximum therapeutic benefits for patients. Moreover, in HER2-positive breast tumors, the cell-surface oncogene HER2 is massively overexpressed, offering two distinct and potential therapeutic approaches. The first involves agents that inhibit signaling functions and the second involves agents designed to deliver tumoricidal effectors to tumor cells. The first approach is more difficult; the only option available for the treatment of breast cancer is lapatinib, whereas the latter approach is transformative. Rastuzumab and pertuzumab are the main options in the management of early-stage breast cancer. Although the development of predictive biomarkers has been challenging and remains under investigation, and the elucidation of immunologic markers has been difficult, these markers remain a matter of ongoing studies. The therapy approaches for various stages of HER2-positive breast cancer are summarized in Table 1.

## 3. Precision-Medicine Approach to Metastatic Breast Cancer Therapy

Precision medicine is an approach to medical treatment that considers the individual differences in people’s genes, environments, and lifestyles. This approach allows for more personalized and effective medical treatments, as it considers the unique characteristics of each patient’s cancer rather than treating all patients with the same therapy.

Metastatic breast cancer (MBC) rapidly extends to other parts of the body, such as the bones, liver, lung, and brain. MBC is a significant public health issue, with an estimated 155,000 women in the US and over 500,000 women worldwide living with the disease [60].

Recent advancements in precision medicine have led to the development of new and effective treatments for metastatic breast cancer. One of the main advancements has involved the use of genomic testing to identify the specific genetic mutations that are driving the cancer’s growth. This information is then used to guide treatment for each individual patient [61]. For example, the use of genomic testing has led to the identification of targetable genetic mutations in metastatic breast cancer, such as BRCA1 and BRCA2 mutations. The use of targeted therapies that specifically target these genetic mutations has been shown to be effective in treating patients with BRCA-positive breast cancers. One example of a targeted therapy for BRCA-positive breast cancer is the use of PARP inhibitors, such as olaparib (Lynparza) [36].

Another example of the use of precision medicine in the treatment of metastatic breast cancer is the use of immunotherapy. Immunotherapy is a type of treatment that helps the body’s immune system to recognize and attack cancer cells. Immune checkpoint inhibitors, such as pembrolizumab (Keytruda), have been approved for the treatment of metastatic breast cancer and have shown promising results in clinical trials [62].

In addition to targeted therapies and immunotherapy, precision medicine has also led to the development of new endocrine therapies for metastatic breast cancer. Endocrine therapy is a type of treatment that targets the hormones that drive the growth of breast cancer cells. New endocrine therapies, such as CDK4/6 inhibitors including palbociclib (Ibrance), have been shown to be effective in treating patients with hormone-receptor-positive metastatic breast cancer [63,64].

Another important aspect of precision medicine in metastatic breast cancer is the utilization of biomarkers to find out which patients are the most predictive responders to a particular treatment. This allows for more accurate treatment selection and improved outcomes, as patients are more likely to receive treatments that are best suited to their specific type of cancer [65].

One major area of precision-medicine progress in metastatic breast cancer is the use of liquid biopsy [66]. A liquid biopsy is a non-invasive test that uses a sample of a patient’s blood to detect cancer cells or genetic information about the cancer. This allows for easier and less invasive monitoring of disease progression, as well as the identification of new mutations that may arise during treatment.

Another area of advancement is the use of precision radiotherapy. Radiotherapy uses high-energy radiation to kill cancer cells and shrink tumors. Precision radiotherapy uses advanced imaging techniques to more precisely target the radiation to the cancerous area, reducing the exposure of healthy tissue to radiation and reducing the risk of side effects [67,68].

Precision medicine is a rapidly evolving field of medicine that aims to provide individualized treatment to patients based on their genetic and molecular characteristics. This approach has shown great promise in the treatment of metastatic breast cancer, which is considered incurable. Despite its potential benefits, precision medicine in MBC faces several limitations to overcome in order to achieve optimal outcomes for patients [69,70].

One of the main limitations of precision medicine in MBC is the lack of availability of actionable targets for therapy [71,72]. MBC is a heterogeneous disease, meaning that it can arise from different mutations in different patients, making it difficult to identify a single target that can be effectively treated in all cases. This means that precision-medicine strategies may not be applicable to all patients with MBC, and further research is needed to identify new targets that can be used to develop new therapies.

Another limitation is the cost of genetic testing and personalized treatment [73]. The cost of genetic testing and the development of personalized therapies is high, which may make it difficult for many patients to access these treatments. This can create a financial burden for patients and their families, as well as for the healthcare system as a whole [74,75]. In order to address this issue, it is important for healthcare providers, insurance companies, and government agencies to work together to ensure that precision-medicine treatments are affordable and accessible to all patients who need them [76].

A third limitation is the limited availability of clinical trials for personalized treatments in MBC [70,77]. Many new treatments in the field of precision medicine are still in the early stages of development and are not yet widely available to patients. This means that many patients with MBC may not have access to the latest treatments and may not be able to participate in clinical trials to test the efficacy of these treatments [78,79]. This can lead to a lack of data on the effectiveness of personalized treatments in MBC, making it difficult to determine their true value in the treatment of this disease.

Another limitation of precision medicine in MBC is the lack of standardization in the molecular testing and treatment approaches [77]. The molecular testing and treatment approaches used for MBC can vary widely from one healthcare provider to another, which can lead to inconsistent results and confusion for patients and healthcare providers. In order to overcome this limitation, it is important for healthcare providers to adopt standardized molecular testing and treatment approaches, and for regulatory agencies to establish clear guidelines for the use of these approaches in MBC.

In order to overcome these limitations, it is important for researchers, healthcare providers, and regulatory agencies to work together to advance the field of precision medicine in MBC. This can involve expanding access to genetic testing, investing in the development of new treatments, and standardizing molecular testing and treatment approaches. In addition, it is important to increase patient awareness of the benefits of precision medicine and to educate patients about the importance of participating in clinical trials to help advance the field [76].

Another solution is to develop new technologies and techniques to improve the accuracy of molecular testing and to make it more widely available to patients. This can involve developing new molecular tests that are more sensitive and specific, as well as new technologies that can be used to analyze large amounts of genetic data in real time. In addition, it is important to invest in the development of new computational tools and algorithms that can help healthcare providers make more informed decisions about personalized treatment for MBC [80,81].

Another way to overcome the limitations of precision medicine in MBC is to increase collaboration and information sharing between researchers and healthcare providers. This can involve creating networks of healthcare providers, researchers, and patients who can share information about the latest treatments and research findings. In addition, it is important to encourage healthcare providers to participate in ongoing education and training programs that focus on recent advancements.

Another solution is to invest in large-scale studies that can provide more comprehensive data on the effectiveness of personalized treatments in MBC. For example, the National Cancer Institute’s Precision Medicine Initiative (PMI) has launched several large-scale studies to investigate the use of genomic analysis in the treatment of cancer [82,83]. These studies will provide valuable data on the benefits and limitations of precision medicine in MBC and will help to inform future treatment decisions for patients.

It is crucial to engage patients and their families in the decision-making process when it comes to precision-medicine treatment for MBC. This can involve educating patients about the benefits and limitations of precision medicine and encouraging them to participate in decision-making and shared decision-making with their healthcare providers. By involving patients in the decision-making process, healthcare providers can ensure that patients receive the best possible treatment for their individual needs [84].

Finally, precision medicine has the potential to revolutionize the treatment of metastatic breast cancer, but there are several limitations that must be overcome in order to achieve optimal outcomes for patients. These limitations include the lack of availability of actionable targets, the cost of genetic testing and personalized treatment, the limited availability of clinical trials, and the lack of standardization in molecular testing and treatment approaches. In order to overcome these limitations, it is important for researchers, healthcare providers, and regulatory agencies to work together to advance the field of precision medicine in MBC, and to involve patients in the decision-making process.

## 4. Current Landscape of Precision-Medicine Therapy for TNBC

It is estimated that 2.3 million new cases of female breast cancer were diagnosed in 2020, representing 11.7% of all new cases [85]. Among them, triple-negative breast cancer (TNBC) is the second most common, affecting between 10 and 20% of diagnosed patients globally [86]. TNBC is a type of breast cancer that does not express hormone (estrogen or progesterone) receptors or human epidermal growth factor receptor 2 (HER2), which represent molecular targets for therapeutic agents [87,88,89]. Therefore, TBNC does not respond to hormonal therapy or HER2 target drugs: the main treatment options which remain are surgery and chemotherapy [14,90].

According to the National Cancer Institute, based on women diagnosed between 2012 and 2018 in the United States, the overall 5-year relative survive rate of TNBC was 77.1%. However, this rate depends on the stage at diagnosis and can be up to 91.3% in tumors localized only in the breast, or 12% in distant tumors that are spread in distant organs as lungs or bones [86]. Another aggravating factor for TNBC patients is its relapses. Approximately one quarter of women with localized disease will experience a relapse with metastasis [91,92], and the highest risk for relapse is in the first 2–3 years after treatment [93]. Several factors can influence the likelihood of a TNBC relapse and overall survival, including the size and stage of the cancer at the time of initial diagnosis, and systemic adjuvant/neoadjuvant chemotherapy.

Surgery is a common treatment option for TNBC. The type of surgery performed will depend on several factors, including the size and location of the cancer, the stage of the disease, and the overall health of the patient. The two main types of surgery for breast cancer are lumpectomy (removal of the cancerous tissue and some surrounding healthy tissue, breast-conserving) and mastectomy (removal of the entire breast, including the tumor and surrounding normal tissue). Patients may be subjected to neoadjuvant or adjuvant radiotherapy, chemotherapy, or their combination, depending on tumor staging [89,90]. A recent review pointed out the benefits of radiotherapy as an adjuvant treatment in advanced TNBC, and it should be evaluated for patients with a high recurrence risk [94].

Besides surgical removal, chemotherapy remains the main treatment for TNBC in neoadjuvant, adjuvant, or metastatic scenarios [87,90,92]. Chemotherapy regimens include cytotoxic agents such as anthracycline (doxorubicin or epirubicin), taxane-based agents (paclitaxel or docetaxel), alkylating agents (cyclophosphamide), antimetabolites (capecitabine), and platinum-based agents (carboplatin or cisplatin) [89,90].

Although surgery and cytotoxic chemotherapy drugs remain the standard of care for TNBC, targeted therapies have been studied for TNBC treatment that show benefits to patients. Several of these agents have received the approval from the US-FDA (Figure 2), including PARP inhibitors, immune checkpoint inhibitors, and antibody–drug conjugates [14]. The treatment choice depends on the specific characteristics of each patient’s cancer, as well as their overall health.

Among targeted therapies, the first to receive FDA approval for TNBC were PARP inhibitors [95]. PARP, or poly ADP ribose polymerase, is an enzyme that helps repair damaged DNA in cells [96,97]. Approximately one quarter of TNBC patients showed a germline BRCA-mutation that prevented them from repairing DNA damage [98]. PARP inhibitors work by blocking the PARP enzyme and preventing the cancer cells from repairing the damage, leading to their death [96,97,99].

Olaparib (Lynparza^®^) was the first PARP inhibitor to receive FDA approval in January 2018 for germline BRCA-mutated, HER2-negative metastatic breast cancer, pre-treated with a neoadjuvant, adjuvant, or metastatic chemotherapy [95]. The approval was based on the OlimpiAD (NCT02000622) clinical trial. This randomized, open-label, phase III clinical trial included 302 patients receiving olaparib as a monotherapy or chemotherapy (capecitabine, eribulin, or vinorelbine), showing that the tested drug was significantly more effective than standard therapy in terms of PFS (7.0 months vs. 4.2 months, respectively). Besides PFS, patients that received olaparib presented fewer grade three adverse events with a lower rate of treatment discontinuation than standard therapy: 4.9% vs. 7.7%, respectively [100]. More recently, the FDA also granted olaparib approval for adjuvant treatment of BRCA-mutated, HER2-negative high-risk early breast cancer, previously treated with chemotherapy in neoadjuvant or adjuvant scenarios, based on the OlympiA (NCT02032823) clinical trial [101]. OlympiA was a phase III, double-blinded clinical trial with 1836 patients randomized for olaparib or placebo treatment. Olaparib showed statistical superiority in invasive disease-free survival and in distant disease-free survival, as primary and secondary objectives, respectively [102].

Talazoparib (Talzenna^®^) is also a PARP inhibitor, which received FDA approval in 2018 based on the EMBRACA trial (NCT01945775), for germline BRCA-mutated, HER2-negative locally advanced or metastatic breast cancer [98]. The EMBRACA trial was a phase III, open-label trial with 431 patients randomized to receive talazoparib or standard therapy (capecitabine, eribulin, gemcitabine, or vinorelbine). Talazoparib-treated patients had significantly better outcomes, with a median progression-free survival of 8.6 months compared to 5.6 months in the placebo arm, and a higher objective response rate (62.6% vs. 27.2%, respectively). Even though grade three and four adverse events occurred more in the talazoparib-treated arm (55% vs. 38%, compared with standard treatment), treatment discontinuation was low (5.9% for talazoparib vs. 8.7% for standard chemotherapy) [103]. A final overall survival analysis did not show significant improvement in overall survival for talazoparib vs. chemotherapy (median OS 19.3 months vs. 19.5 months, respectively), but patient-reported outcomes continued to favor talazoparib [104].

Other PARP inhibitors such as rucaparib, veliparib, and niraparib have also been studied in clinical trials for TNBC but have not yet received FDA approval. Recent reviews have suggested that monotherapy with PARP inhibitors can be effective for BRCA-deficient patients but should be used in combination with other cytotoxic drugs or immunotherapy in the case of BRCA-proficient tumors [99,105,106].

Another target therapy that may benefit TNBC patients is immunotherapy that works by stimulating the immune system to recognize and combat cancer cells. The importance of the immune system in cancer disease course has long been described, with many studies pointing to favorable outcomes in tumors with tumor-infiltrating lymphocytes in the tissue [107,108,109]. Approximately 20% of TNBCs express the programmed death-ligand 1 (PD-L1) molecule [110,111].

In the beginning of 2019, the FDA granted accelerated approval to atezolizumab (Tecentriq^®^) in combination with albumin-bounded paclitaxel (nab-paclitaxel), bringing TNBC into the immunotherapy era. Atezolizumab is a humanized monoclonal antibody against PD-L1 [112]. Approval was granted based on initial results from the IMpassion130 (NCT02425891) clinical trial for unresectable locally advanced or metastatic TNBC with PD-L1 expression [30]. Initial data from the clinical trial showed a prolonged progression-free survival in the atezolizumab + nab-paclitaxel group when compared to the placebo + nab-paclitaxel, both in the intention-to-treat population and the PD-L1-positive subgroup [113,114]. However, a final overall survival analysis failed to prove the benefit of atezolizumab + nab-paclitaxel in the intention-to-treat population, which did not allow for formal analysis in the PD-L1 subgroup [115]. Researchers conducted an exploratory analysis in the subgroup, which showed a higher overall survival in the atezolizumab group (25.4 months vs. 17.9 months). Even so, the FDA and the pharmaceutical company decided on the withdrawal of the U.S. accelerated approval [116].

Pembrolizumab (Keytruda^®^) was the next immunotherapy drug to receive an FDA accelerated approval when used in combination with chemotherapy for locally recurrent unresectable or metastatic TNBC with PD-L1 expression. Pembrolizumab as a monotherapy did not enhance overall survival (OS) nor did PFS when compared with single-agent chemotherapy, but lower grade 3–4 adverse events were observed in the immunotherapy group (14% vs. 36%) [117]. This motivated the investigation of pembrolizumab in combination with chemotherapy during the Keynote-355 clinical trial (NCT02819518). In this trial, 847 patients were double-blinded randomized in two arms: pembrolizumab–chemotherapy (nab-paclitaxel, paclitaxel, or gemcitabine plus carboplatin) or placebo–chemotherapy. Among patients with a combined positive score (CPS) ≥ 10, pembrolizumab–chemotherapy significantly increased the median progression-free survival compared with the placebo (9.7 months vs. 5.6 months), but the same was not observed in the CPS ≥ 1 subgroup or in the intention-to-treat population [118]. A final overall survival analysis reported a statistically significant median overall survival difference in the CPS ≥ 10 subgroup for the pembrolizumab and placebo arms (23.0 months vs. 16.1 months, respectively). No significant difference in overall survival was reported for the CPS ≥ 1 subgroup or intention-to-treat population [119].

More recently, in 2021, the accelerated approval for pembrolizumab became a routine approval, and the FDA also granted pembrolizumab approval as a neoadjuvant for high-risk, early-stage non-metastatic TNBC in combination with chemotherapy and as a monotherapy in adjuvant treatment based on a Keynote-522 (NCT03036488) trial [120,121]. In the neoadjuvant phase, patients received pembrolizumab or a placebo and four cycles of paclitaxel + carboplatin, followed by four cycles of doxorubicin (or epirubicin) + cyclophosphamide, while in the adjuvant phase, patients received pembrolizumab or placebo for nine cycles or until unacceptable toxicity. This regimen of pembrolizumab is the first immunotherapy-based approval for early TNBC [121]. The primary endpoints of Keynote-522 were the pathological complete response rate at surgery and event-free survival in the intention-to-treat population. In the first endpoint, pembrolizumab showed a significant improvement when compared to placebo (64.8% vs. 51.2%, respectively) [120] and a significant improvement in event-free survival (84.3% vs. 76.8%) [122].

The last class to receive US-FDA approval for TNBC was the ADCs. ADCs combine monoclonal antibodies with high-potency cytotoxic small molecules. Sacituzumab govitecan (Trodelvy^®^) received approval for patients with metastatic disease cancer who have received at least two prior therapies for it, based on the IMMU-132-01 (NCT 01631552) trial [123]. One year after accelerated approval, sacituzumab govitecan received regular approval for patients with unresectable locally advanced or metastatic TNBC who have received two or more systemic therapies and one of these for metastatic disease, based on the ASCENT trial (NCT02574455) [124]. Sacituzumab is a humanized antitrophoblast cell-surface antigen 2 (Trop-2) monoclonal antibody, and govitecan is a topoisomerase I inhibitor [125,126]. Trop-2 is a transmembrane calcium signal transducer, overexpressed in tumors, associated with tumor growth, invasion, and spread [126,127,128]. During the IMMU-132-01, a phase I/II trial, sacituzumab govitecan was associated with durable objective responses in refractory metastatic TNBC, with myelotoxicity as the main adverse effect observed [125]. Later, the phase III ASCENT trial showed improvements in median progression-free survival (5.6 months vs. 1.7 months, *p* < 0.001) and median overall survival (12.1 months vs. 6.7 months, *p* < 0.001) for the sacituzumab govitecan arm when compared to single-agent chemotherapy (eribulin, vinorelbine, capecitabine, or gemcitabine), respectively [129]. Other drug-conjugates have been studied for TNBC, such as ladiratuzumab vedotin, U3-1402, not yet approved for clinical use [130].

Despite the significant and promising results shown in clinical trials for immunotherapy, some reviews have pointed out that PD-L1 is not the ideal biomarker to be used in patients’ selection for anti-PD-L1/anti-PD-1 therapies and the prediction of responses to immunotherapy [131,132,133]. Other biomarkers are currently under investigation, such as tumor mutational burden, the presence of tumor-infiltrating lymphocytes, microsatellite instability, LDH levels, the presence of visceral disease, major histocompatibility complex, the detection of circulating tumor DNA, the value of CD274 amplifications, and immune gene expression profiles [131,132,133,134]. Despite of the search for new and better biomarkers, other forms of immunotherapy are also currently under investigation, which include vaccines, adoptive cell therapies, autologous tumor-infiltrating lymphocytes, oncolytic virus therapies, and cytokine agents [14,133,135].

## 5. Potential Applications of Precision-Medicine Therapy for TNBC

TNBC represents a subset of breast cancers with limited treatment options due to their lacking ER, PR, and HER2. Around 10 to 20% of newly diagnosed breast cancer cases are TNBC, which is associated with the incidence of visceral metastases, a high risk of early-recurrence, and a poor prognosis [87,136,137]. In TNBC signaling, receptor tyrosine kinases (RTKs) function through two major downstream cascades, RAS/MAPK and PI3K/AKT/mTOR. In TNBC cells, RTKs transduce signals downstream of EGFR, platelet-derived growth factor receptor (PDEGFR), vascular endothelial growth factor receptor (VEGFR), insulin-like growth factor receptor (IGFR), fibroblast growth factor receptor (FGFR), and transforming growth factor beta (TGF-β). EGFR expression is associated with the aggressiveness of TNBC tumors. Almost 60 to 80% TNBC tumors have dysregulated EGFR expression [138]. Post neoadjuvant therapy, EGFR expression persists frequently in TNBC. Therefore, ant-EGFR therapy may be effective in patients with therapy-refractory EGFR-positive TNBC tumors [139]. The KRAS/SIAH/EGFR pathway is frequently upregulated in TNBC (SIAH, seven absentia homolog) [139,140]. Paired gene expression of EGFR and SIAH may be a prognostic biomarker in TNBC. A reduction in EGFR and SIAH levels may be utilized to predict therapy benefits. EGFR-targeted anticancer therapy is attractive for TNBC. However, this therapy approach may be associated with off-target effects [141]. In a clinical trial, the anti-EGFR cetuximab demonstrated an unsatisfactory outcome due to the stimulation of a compensatory signaling pathway, PI3K/AKT [140]. A dual EGFR/HER2 RTKi, effective in HER2+ breast cancer, was however ineffective in TNBC [142]. The MEK inhibitor selumetinib inhibited invasiveness in MDA-MB-231 and SUM149 TNC cell lines in vitro [138]. Selumetinib reduced lung metastasis in a TNBC-bearing mouse xenograft model. Further, the combination of MEK inhibitors with PD-L1/PD-1 inhibitors enhanced therapeutic efficiency in a murine syngeneic TNBC model [14,143].

Dysregulation of the PI3K/AKT/mTOR pathway often occurs in TNBC. AKT and mTOR hyperactivation indicate a poor prognosis in TNBC patients. Therefore, the double hindrance of AKT and mTOR may be a promising therapy approach in TNBC [138,144]. AKT inhibitors, such as ipatasertib and capivasertib, have shown beneficial results in aggressive TNBC [145,146]. The inhibition of AKT or mTOR can sensitize resistant cells to cis-platin-induced apoptosis [147]. Everolimus in combination with carboplatin is an efficient therapy for metastatic TNBC [148,149]. PI3K inhibitors may downregulate BRCA1 and BRCA2 expression and sensitize BRCA-proficient TNBC tumors to PARPi [150]. Double PI3K and PARP inhibition using buparlisib (BKM120) and olaparib substantially decreased the proliferation of the BRCA-proficient TNBC cell lines MDA-MB-231 and MDA231-LM2 [151]. A clinical trial with buparlisib (BKM120) and olaparib is ongoing [138]. Resistance to mTOR inhibitor in TNBC was associated with the presence of Notch-dependent cancer-stem-like cells (CSCs) [152]. As a result, Notch inhibitors may be combined with AKT or mTOR inhibitors for effective TNBC therapy [14].

Notch signaling stimulation is associated with TNBC tumor growth, expansion, CSCs development, tumor invasiveness, and metastasis development [153,154,155,156]. There have been many approaches to develop Notch inhibitors for TNBC. Several Notch inhibitors are currently in different phases of clinical trials for TNBC therapy [156]. Numerous pre-clinical studies have indicated that gamma-secretase-inhibitors (GSIs) are promising for the inhibition of TNBC tumors. However, GSI therapies for TNBC are associated with dose-dependent and mechanism-based gastrointestinal (GI) toxicities. Therefore, an intermittent dosing of GSI therapies may be utilized for TNBC therapy with low toxicities [156,157]. Further, non-GSI and a new generation of Notch inhibitors such as CB-103 are particularly promising for TNBC therapy.

Cyclin-dependent kinases (CDKs) such as CDK4/CDK6 overexpression are common in different cancers including TNBC. Based on pathological cyclin D1-associated kinase activity, such as in cancer, CDK4/CDK6 expression can be deranged, overexpressed, or altered. CDK4/CDK6 serves as major node of integration downstream of multiple signaling pathways, in which their stimulation prompts progression into the cell cycle. CDK4/CDK6 can phosphorylate retinoblastoma (RB) protein and the related proteins p107 and p130. RB proteins have functions in cell cycle regulation. Therefore, the CDK4/CDK6–RB axis is crucial to cell-cycle entry, and the vast majority of cancers destabilize this axis to promote proliferation. The deregulation of the CDK4/CDK6–RB axis is the direct oncogenic activation of CDK4/CDK6 function. CDK4/CDK6, as CDK proteins, are exposed to phosphoregulation. Cyclin D1 is unstable and actively shuttles between the cytoplasm and nucleus. Deregulated cyclin D1 protein expression, gene translocation, and gene amplification are detected in many cancer types. Deregulated cyclin D1 is frequent in breast cancer [158]. CDK inhibitors have been utilized in TNBC therapy with promising outcomes. The CDK inhibitor dinaciclib is in a phase I clinical trial for metastatic or advanced breast cancer and TNBC therapy. A CDK4/6 inhibitor, trilaciclib, in combination with gemcitabine is in a phase I clinical trial with mTNBC. Another CDK4/6 inhibitor, ribociclib, in combination with bicalutamide (an androgen receptor inhibitor, ARi) is in phase I/II clinical trials for advanced AR-positive TNBC patients. AR expression is upregulated in 10 to 43% of TNBCs. Treatment with AR antagonists has demonstrated clinical benefits in TNBC patients. An AR antagonist, darolutamide, is in an ongoing phase II clinical trial for the treatment of unresectable or metastatic TNBC [140,159,160,161,162]. Further, the VEGF inhibitor bevacizumab in combination with chemotherapy has been shown to improve PFS for TNBC patients compared to chemotherapy alone [140,163].

Lyotropic liquid crystalline lipid nanoparticles consisting of an internal cubic phase nanostructure are known as cubosomes. Recently, cubosomes have attracted attention as an innovative drug delivery agent for cancer therapy [164,165,166,167]. Cubosomes have many benefits over liposomes including increased stability, enhanced drug loading competence, and the ability to encapsulate both hydrophobic and hydrophilic drugs. A recent study has demonstrated the active targeting of cancer cells by drug-loaded cubosomes for cancer therapy. In this study, a cubosome was fabricated loading the anticancer drug copper-acetaylacetonato and functionalizing with hyaluronic acid (HA), the ligand for cell-surface receptor CD44. CD44 is overexpressed in different cancer cells including TNBC cells. HA-conjugated, copper-acetylacetonato-loaded NPs with a 152-nm hydrodynamic diameter were effectively taken up by a TNBC cell line, MDA-MD-231; however, they were not taken up by MCF-7 cells (with no CD44 expression). In the analysis, HA-functionalized cubosomes targeting CD44 inhibited TNBC cell lines more effectively than non-targeted cubosomes in CD44-positive cells, indicating the significance of targeted therapy utilizing cubosomes. Specific targeting and apoptotic cell death were evident in 2D-culture and 3D-spheroids studies. This study demonstrated that HA-conjugated, copper-acetylacetonato-loaded cubosomes are promising for the selective targeting of CD44-expressd tumors and may be an effective therapeutic approach for TNBC. Further, carcinoembryonic antigen (CEA) is a cell-surface glycoprotein overexpressed in 50% of breast cancers. Therefore, engineered cubosomes may be utilized in CEA-targeted therapy of breast cancer [166,167].

An aptamer-guided siRNA–NP targeting CD44 expression in TNBC was formulated by Alshaer et al. [168]. The core of the NPs was made up of siRNA–protamine complex, and the shell contained an aptamer ligand for targeting CD44 expressed on TNBC cells. The outcome of the study indicated that the formulation targeting TNBC cells demonstrated enhanced anticancer effects [168,169].

ADCs are a potential and promising targeted therapeutic approach for breast cancer and TNBC. In 2020, sacituzumab govitecan gained accelerated approval from the US-FDA for metastatic TNBC patients who have received at least two prior therapies for metastatic disease. In 2021, the US-FDA finally provided full approval for sacituzumab govitecan (Trodelvy by Immunomedics Inc) for patients with unresectable locally progressed or metastatic TNBC who have received two or more prior systemic therapies—at least one of these for metastatic disease. As mentioned previously, sacituzumab govitecan is composed of an antibody coupled to topoisomerase I inhibitor (SN-38) through a proprietary hydrolysable linker, targeting human trophoblast cell-surface antigen 2 (Trop-2) which is expressed in the majority of breast cancers including TNBC [123,124,125,126,127,128]. ADCs with highly cytotoxic drugs represent a prospective targeted therapy approach for TNBC patients. Further improvement in payload, linker chemistry, and ADC drug release mechanisms will enhance TNBC therapy for the better survival of patients.

Oncolytic viruses spread throughout tumors and promote antitumor responses by dual mechanisms: cancer cell inhibition and the initiation of anticancer immunity [170]. When injected into tumors, cell debris and antigens are released by oncolytic viral infections to activate the immune system [171]. Thus, the subtype of TNBCs susceptible to ICIs may be sensitized by OV delivery. Non-immunogenic tumors can be made prone to ICIs by OVs. Thus, OVs may be utilized to enhance ICI-based treatment for TNBC patients. Clinical trials of TNBC utilizing oncolytic virotherapy are currently in different phases of evaluation [135].

TVEC (talimogene laherparepvec) is a modified herpes simplex 1 virus that includes coding sequences of protein GM-CSF, which may stimulate the immune system when administered into tumors. When injected directly into tumors, it undergoes replication within the tumor cells, resulting in the breakdown of the tumor cells and the production of tumor-derived antigens. Then, immune cells can identify the antigens and penetrate the tumors and target tumor cells to inhibit them. OV could be effective in combination with chemotherapy when delivered to TNBC tumors as a neoadjuvant therapy. In a phase II trial with 37 patients, 45.9% of patients demonstrated a response, 89% remained disease-free two years post-therapy, and no disease relapse was detected in patients showing a potential response to OV-based combination therapy [172].

A phase Ib/II trial based on subtyping and guided by a genome biomarker evaluated the efficiency of TNBC therapy in seven arms: pyrotinib with capecitabine (A), androgen receptor inhibitor with CDK4/6 inhibitors (B), anti PD-1 with nab-paclitaxel (C), PARP inhibitor included (D) and anti-VEGFR included (E), anti-VEGFR included (F), and mTOR inhibitor with paclitaxel (G) (Figure 3) in refractory metastatic TNBC patients with a median of three previous lines of therapy [136]. The result indicated that the immunotherapy arm, antiPD-1 with nab-paclitaxel, achieved the highest objective response rate (ORR) (complete response 52.6%+, partial response 95%, confidence interval (CI) 18.7–41.2%). This study demonstrated the clinical benefit of subtyping-based targeted therapy for refractory metastatic TNBC. The combination of subtyping and genomic sequencing can be utilized to screen patients to select for the targeted therapy [136].

TNBC is a largely fatal form of breast cancer due to drug resistance. TNBC tumors may be sensitive to GLUT1 (glucose transporter 1) inhibition due to the high expression of GLUT1 and metabolic dependency of TNBC cells [173]. A recent study reported that the inhibition of GLUT1 by BAY-876 impeded the growth of a subset of TNBCs exhibiting enhanced glycolytic and lower oxidative phosphorylation (OXPHOS) rates. The study found that sensitivity to GLUT1 inhibition is potentially allied with the level of retinoblastoma tumor suppressor protein 1 (RB1). RB1-negative cells are insensitive to GLUT1 inhibition. An examination of breast cancer patients exposed heterogeneous RB1 expression at both the mRNA and protein levels in basal tumors. Around 74% of basal-like breast tumors are RB1-positive and may be sensitive to GLUT1 inhibition. Around 20 to 30% of basal tumors have a low expression of RB1 and show less sensitivity to GLUT1 inhibitors [12,174,175]. In the RB1-low subtype of TNBC, RB1 loss activates mitochondrial biogenesis with enhanced OXPHOS rates [174,176]. RB1 expression is inversely correlated with the increased expression of OXPHOS genes in TNBC tumors. As a result, drugs targeting mitochondrial functions may also exert anti-tumor effects in RB1-low TNBC tumors [174,176]. The study emphasized the role of the RB1 tumor suppressor protein as a key regulator of cell metabolism and determinant of TNBCs’ sensitivity to the metabolic inhibitor BAY-876. The study demonstrated an RB1-protein-dependent metabolic addiction to GLUT1 function in a subset of TNBC tumors, recognizing BAY-876 as an efficient agent to inhibit the growth of TNBC cells in patient-derived animal models that express the RB1 protein. This study suggests a consideration of RB1 protein level heterogeneity in the development of personalized metabolic therapeutic strategies toward TNBC therapy. The differing levels of RB1 may be used as a biomarker to discriminate treatment-responder and non-responder TNBC patients. As a result, a better understanding of the molecular complexity of cancer cells may enhance the opportunity for targeting with precision. Building up an enriched pharmacy of cancer drugs matching a specific change in the cancer cells may improve the chance of cure [173].

Metabolic reprogramming is a significant hallmark of cancer. Systemic characterization of metabolites in TNBC may be useful in therapeutic targeting in TNBC. Tumor-promoting metabolites may be a prospective target in TNBC therapy [177]. One study reported the polar metabolome and lipidome in 330 TNBC samples and 149 paired normal breast tissues to construct a metabolomic atlas of TNBC. Combining the previously established transcriptomic and genomic data of the same cohort, the study linked the TNBC metabolome to the genome by comprehensive analysis and classified TNBCs into three distinct metabolomic subgroups (C1 to C3). C1 demonstrated an enrichment of ceramides and fatty acids; C2 featured metabolites related to oxidation reactions and glycosyl transfer, and C3 was characterized by the lowest level of metabolic reprogramming. Based on the developed metabolomic dataset and optimizing previous transcriptomic subtypes, the study identified some important subtype-specific metabolites as potential therapeutic targets for TNBC patients. According to the study, the transcriptomic luminal androgen receptor (LAR) subtype overlaps with the metabolomic C1 subtype. Patient-derived organoid and xenograft models have demonstrated that an intermediate of ceramide may be utilized: sphingosine-1-phosphate (S1P)-targeted therapy for the LAR subtype is prospective. Further, basal-like immune-suppressed (BLIS) tumors contained two metabolomic subtypes, C2 and C3. N-acetyl-aspartyl-glutamate is a significant tumor-promoting metabolite and a crucial target for the BLIS subtype [177].

TNBC is a highly heterogeneous disease with frequent multi-clonality. Therefore, precision combination therapies may be prospective for TNBC therapy. Combination therapy of targeted agents with chemotherapy or other targeted agents may be an effective therapy approach. Combination therapy of the PARPi olaparib with PI3Ki buparlisib and carboplatin resulted in the inhibition of TNBC tumors [178]. A phase I study of olaparib–buparlisib combination therapy for TNBC is under assessment (NCT01623349). PARPi in combination with ADCs and chemotherapy has demonstrated an enhanced inhibition of TNBC cells. As such, combination therapy of olaparib with sacituzumab govitecan was effective against TNBC tumors with or without BRCA mutations [179]. Further, iniparib in combination with gemcitabine and carboplatin inhibited TNBC cells [180]. PARPi, when utilized in combination with PD-L1 inhibitors, could sensitize the PARPi against TNBC cells. Buparlisib in combination with disulfiram/Cu (DSF/Cu) and paclitaxel resulted in a reduced tumor burden and disease relapse rate in TNBC compared to paclitaxel alone [181]. A clinical trial including the AKT inhibitor ipatasertib in combination with paclitaxel showed a promising outcome against TNBC (NCT02162719). A clinical trial with the GSI PF-03084014 combined with the AKT inhibitor MK-2206 or NF-kβ inhibitor Bay11-7082 may be effective against TNBC with a Notch mutation and wild-type PTEN [182]. In another study, a Notch3 inhibitor increased the efficiency of the TKI gefitinib targeting EGFR in TNBC cells. Since TNBC is a highly heterogeneous deadly disease, the development of potential therapy strategies necessitates further investigations for monotherapy, as well as combination therapy, to reduce mortality and improve the survival of TNBC patients.

## 6. Clinical Trials Status in Metastatic BC and TNBC

### 6.1. PI3K-AKT-mTOR Axis Inhibitors

Individual patient care is tailored to specific patients’ genetic data, and environmental and lifestyle information in clinical settings is critical to several trials edging closer to approval. The better prognosis of breast cancer has paved the way toward the selection of potential molecular targets of therapeutics and trials involving the mutational status of these agents (Figure 4). Deregulation of the PI3K, AKT, and mTOR axis pathway is a major contributor toward the development of tumors, and inhibitor drugs are under experiment. Buparlisib, for example, is undergoing phase III clinical trials in postmenopausal HR+ and HER2–breast cancer. Patients in 29 countries (267 treatment centers) were enrolled with a known PI3K pathway activated or non-activated status in order to test treatment efficacy and PFS. In another multicenter clinical trial with this drug, metastatic breast cancer-relapsed patients after treatment with endocrine or mTOR inhibitors were recruited. The findings of these trials are summarized in Table 2.

Everolimus is another inhibitor of mTOR, approved worldwide from clinical trials to clinical practices for the treatment of breast cancer in combination with exemestane (an aromatase inhibitor). The efficacy of everolimus in combination with exemestane has been tested in a phase III clinical study (BOLERO-2), in 724 patients with HR+ advanced breast cancer after non-steroidal aromatase inhibitor therapy, where it was associated with a markedly improved median PFS (7.8 months) versus a placebo plus exemestane (3.2 months). In order to analyze the genetic correlations, an exploration of the genetic landscape was revealed by next-generation sequencing. The advantage in PFS when utilizing everolimus was reliable for EGFR1, CCND1, and PIK3CA [19,183,184].

Ipatasertib, an AKT inhibitor drug, in combination with paclitaxel versus a placebo (median PFS was 6.2 and 4.9 months respectively) was evaluated in a phase II clinical study (LOTUS) conducted after the recruitment of 124 patients with locally advanced or metastatic TNBC and 48 patients with PTEN-low tumors (median PFS was 6.2 and 3.7 months, respectively). Likewise, the AKT inhibitor drug AZD5363 in patients with AKT1 E17K-mutant ER+ breast cancer and MK-2206 in patients with PIK3CA-mutant ER+ and HER2– breast cancer was investigated in neoadjuvant settings for individualized treatment options. The investigation of AKT1 E17K as a therapeutic target in ER-positive breast cancer was carried out after treatment with AZD5363 (an ATP-competitive AKT kinase inhibitor). The clinical data suggested (the median PFS was 5.5 months) a strong signal of activity; however, drug combinations may be required to realize the full potential of AZD5363. The combined endocrine treatment with AKT inhibition induced apoptosis in preclinical settings. The anti-proliferative activity of MK-2206, an AKT inhibitor, in multiple human cancer cell lines—and the evidence of human toleration and AKT inhibition—led the way towards investigation in patients with PIK3CA-mutant ER-positive and HER2-negative breast cancers in neoadjuvant settings for individualized treatment options. The pathologic complete response was not observed and the efficacy of endocrine therapeutics (anastrozole) in combination with MK-2206 was unlikely to have been more effective than either alone in phase II trials, limiting further studies [20,21,185].

### 6.2. RAS-RAF-MAPK Inhibitors

RAS-RAF-MAPK is a complex signal transduction pathway from cell-surface receptors to DNA in the nucleus that is directly involved in the formation of new blood vessels after the expression of various genes, with RAS being the most frequently mutated gene. The safety and efficacy of sorafenib (a RAF-1 inhibitor) in metastatic HER2-negative breast cancer was not much appreciated in phase III clinical trial data (RESILIENCE), as it failed to prolong PFS. Mitogen-activated protein kinases (MAPK) inhibitors such as ralimetinib have exhibited acceptable pharmacokinetic characteristics in phase II clinical studies. p38 MAPK is activated by cytokines, ultraviolet irradiation, and/or stress, having an important role in cell survival and the regulation of cytokines [1,186,187]. The clinical development of ulixertinib, a potent and selective inhibitor of extracellular signal regulated kinases (ERK), is also underway for the treatment of multidrug resistance due to reactivation of ERK signaling [188].

### 6.3. Histone Deacetylase Inhibitors

The compact form of DNA called the nucleosome is organized after bonding with acetylated histone units that are involved in the transcription process. Deacetylation is involved in regaining the positive charge of histones that increases the affinity of DNA that represses gene expression after tight wrapping of DNA. The altered gene expression, epigenetic, is responsible for uncontrolled cell proliferation and resistance mechanisms. HDACis cause hyper-acetylation of histones in order to activate the apoptosis and autophagy of cancer cells depending upon their genetic predispositions. YCW1, a HDACi, has shown the autophagic cell death of TNBC cells in preclinical studies after protein down-regulation [189,190]. Entinostat is a selective HDAC inhibitor oral agent, successfully tested for ER+ breast cancer in a phase II randomized clinical trial (ENCORE301), with an improved median PFS of 4.3 and 2.3 months in combination and with exemestane alone [15]. Phase III clinical trials are also underway for further insights into the personalized therapeutic potential of this compound [1,24,191,192].

### 6.4. Miscellaneous

The activation of PARP plays a key role in the development and progression of cancer cells, and the cytotoxicity of chemotherapeutics may be improved by combinatorial strategies with PARPi. Iniparib was the first PARPi inhibitor tested for its efficacy in stage IV recurrent TNBC breast cancer in phase III clinical trials. Olaparib underwent phase III clinical trials (OlympiaD) in comparison to standard chemotherapy in patients with gBRCA-mutated HER2− metastatic breast cancer: olaparib significantly improved the PFS (7.0 months) in these patients. Recently, talazoparib has been evaluated in a randomized phase III clinical study (EMBRACA) in patients with progressed breast cancer, exhibiting significant improvement in PFS (8.6 months) when compared with placebo (5.6 months) [31,33,193].

The cyclin-dependent kinases inhibitors, CDKis (palbociclib, ibociclib, abemaciclib), cause cell-cycle arrest in the G1 phase of cell division and the clinical development of these agents is underway [194,195].

ADCs are chemically modified monoclonal antibodies that, in combination with potential cytotoxic agents, manifest as flourishing targeted therapeutic agents for the treatment of TNBC. Glembatumumab vedotin is an important ADC agent conjugated with a potent cytotoxic agent. Auristatin E was tested in a randomized phase II clinical trial (EMERGE). In recent years, FDA approval has been granted to certain ADC drugs for the treatment of BC and TNBC, which may be a potential approach for the treatment of BC and TNBC in the clinic (Table 3) [14,50,127,128,129,129,130,131,132,196,197].

## 7. Conclusions and Future Perspective

Precision medicine is a prospective approach to lower the off-target toxicities of chemotherapeutic agents and enhance the benefits to patients in the clinic. This is a crucial approach for the effective treatment of breast cancer. Precision medicine requires the selection of suitable biomarkers to predict the efficiency of targeted therapy in specific groups of patients. In breast cancer therapy, several possible druggable mutations have been identified in breast tumors that may be utilized for therapy design. Developments in omics technologies have focused on the more precise strategies of precision therapy. The development of next-generation sequencing technologies has raised hopes for precision-medicine treatment strategies in breast cancer and triple-negative breast cancer. Targeted therapy approaches such as ICIs, EGFRi, PARPi, ADCs, CD44i, OVs, and GLUT1i are prospective treatment options for BC and TNBC. Further, targeting signaling pathways may also represent a promising approach for breast cancer therapy. Combination therapy strategies in combination with appropriate single modalities may significantly improve the precision-medicine treatment of metastatic breast cancer and TNBC patients. However, the development of predictive biomarkers and immunologic markers has been challenging and necessitates further investigation. The development and utilization of next-generation sequencing technologies, and further clinical trials on precision-medicine-based therapeutics, will enhance breast cancer therapy and improve the survival of patients. Precision-medicine approaches have the potential to revolutionize the therapy of metastatic breast cancer and TNBC. It is crucial for researchers, healthcare providers, and regulatory bodies to work in an organized way to advance the field of precision medicine for deadly breast cancer therapy.

## Figures and Tables

**Figure 1 cancers-15-02204-f001:**
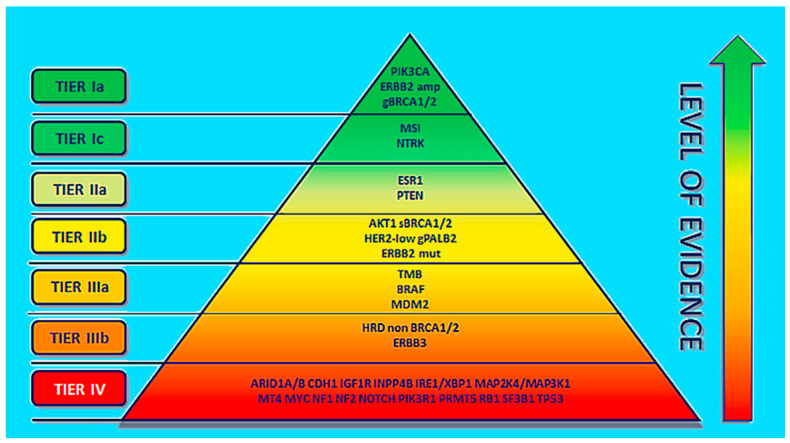
ESCAT ranking of molecular alterations in a specific cancer type (adapted from ref. [3]).

**Figure 2 cancers-15-02204-f002:**
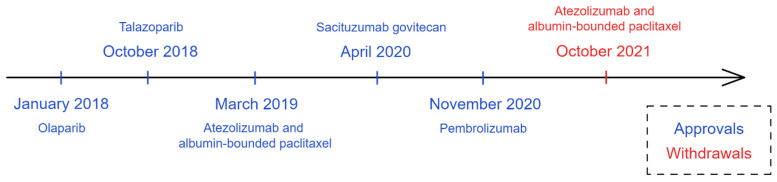
Timeline of US-FDA targeted therapy approvals and withdrawals for triple-negative breast cancer treatment. In blue, drugs approved and in red, drugs withdrawn.

**Figure 3 cancers-15-02204-f003:**
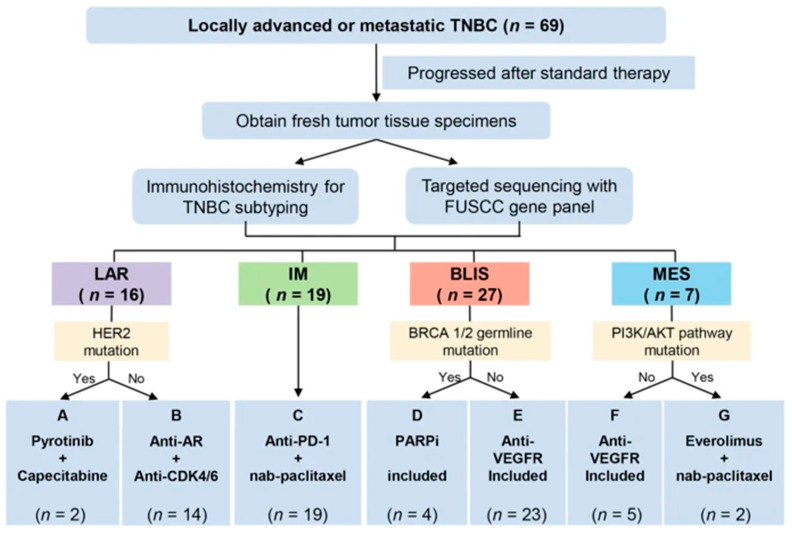
The FUTURE trial schema: integrating TNBC subtyping and genomic targeting. n, number of the patients; TNBC, triple-negative breast cancer; FUSCC, Fudan University Shanghai Cancer Center; LAR, luminal androgen receptor; IM, immunomodulatory; BLIS, basal-like immune-suppressed; MES, mesenchymal-like; AR, androgen receptor; PD-1, programmed cell death-1; PARPi, poly ADP-ribose polymerase inhibitor; VEGFR, vascular endothelial growth factor receptor (adapted from [136]).

**Figure 4 cancers-15-02204-f004:**
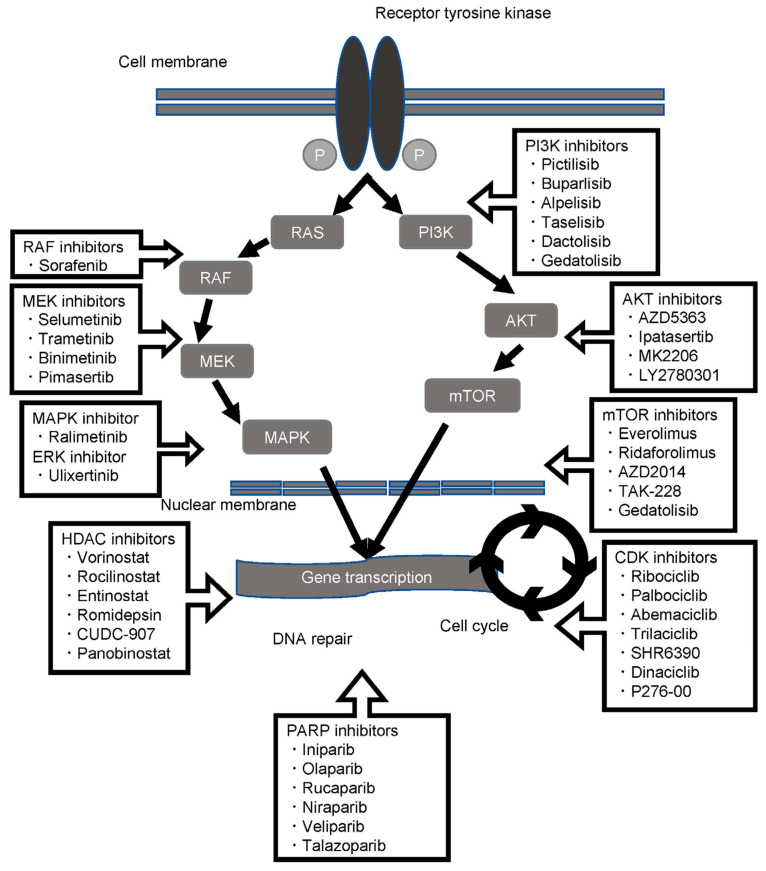
Different molecular pathways and potential targets for breast cancer therapy. Adapted from ref. [1].

**Table 1 cancers-15-02204-t001:** Treatment protocols for various stages of HER2-positive breast cancer.

Breast Cancer Description	Treatment Protocol	Reference
Early-stage breast cancer (adjuvant/neoadjuvant)	Chemotherapy, Trastuzumab, Pertuzumab	[49]
Metastatic breast cancer		
First-line therapy	Taxane, trastuzumab, pertuzumab	[50,51]
Second-line therapy	Trastuzumab emtansine	[52,53]
Third-line therapy	Capecitabine, lapatinib	[54]
	Aromatase inhibitor Trastuzumab/lapatinib	[55]
	Chemotherapy, trastuzumab	[56]
	Lapatinib, trastuzumab	[57]
	Vinorelbine, trastuzumab	[58,59]

**Table 2 cancers-15-02204-t002:** Brief findings of clinical trial number NCT01610284 [16] and NCT01633060 [17].

Total Patient Population (n)	Median PFS (Months)	
	Placebo Group	Buparlisib Group
n = 1147	5.0	6.9
n = 851 (with known PI3K status)	4.5	6.8
n = 372 (PI3K pathway activated status)	4.0	6.8
n = 432 (pre-treatment with endocrine therapy or mTOR inhibitors)	1.8	3.9

**Table 3 cancers-15-02204-t003:** Clinical trial status of potential targets of TNBC.

Target Molecule/Pathway	Chemical Agent	Clinical Trials	Phase/Status of Clinical Trial	References
EGFR	Afatinib	NCT02511847	II	[198,199]
	Gefitinib	NCT01732276	II	
	Panitumumab	NCT00894504	II	
VEGFR	Bevacizumab	NCT01898117	II	[200,201,202]
	Apatnib	NCT01176669	II	
	Cediranib maleate	NCT01116648	I/II	
HGFR/c-MET	Tivantinib	NCT01575522	II	[203]
PTKs	Cabozantinib	NCT01738438	II	[204,205]
	Lucitanib	NCT02202746	II	
PI3K/AKT/mTOR pathway	BKM120	NCT02000882	II	[206,207,208,209,210,211,212]
	BKM120	NCT01629615	II	
	BKM120/BYL719	Nct01623349	I	
	Taselisib	NCT02457910	I/II	
	AZD8186	NCT01884285	I	
	ARQ092	NCT02476955	I	
	AZD5363	NCT02423603	II	
	MK2206	NCT01319539	II	
	Ipatasertib	NCT02162719	II	
	Everolimus	NCT02616848	I	
	Temsirolimus	NCT01111825	II	
AR	GTx-024	NCT02368691	II	[213,214,215]
	Bicalutamide	NCT02348281	II	
	Enzalutamide	NCT02689427	II	
BRCA mutation	Rucaparib	NCT01074970	II	[100,216,217,218,219]
	E7449	NCT01618136	I/II	
	Iniparib	NCT01045304	II	
	Iniparib	NCT01204125	II	
	Veliparib	NCT01306032	II	
	Talazoparib	NCT02627430	I	
PD-1	PDR001	NCT02404441	I/II	[220,221]
	Pembrolizumab	NCT02452424	I/II	
	Pembrolizumab	NCT02657889	I/II	
	Durvalumab	NCT02628132	I/II	

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
