# Peer review of "Recent Advances with Precision Medicine Treatment for Breast Cancer including Triple-Negative Sub-Type"

_cancers, 2023, doi:10.3390/cancers15082204_

Round 1
Reviewer 1 Report
The authors have written a well summarized review on precision and targeted therapeutics for breast cancer subtypes including the TNBC.
The review covers good information for Her2 based targeted therapeutics and its current challenges. Also including the various immune check point inhibitors such as the PI3K/AKT/mTOR. The review also mentions the PD-L1 and EGFR targeting.
However the authors could also include some literature for CD44 receptor and how this could be used for precision based medicines for TNBC and breast cancers include the breast cancer stem cells. CD44 receptor has been widely used for targeted therapeutics in TNBC using nanomedicine (https://doi.org/10.1021/acs.molpharmaceut.2c00439).
Similarly nanomedicines particularly the cubosome nanocarriers have been used for precision based therapy against the carcinoembryonic based antigen (CEA). CEA is also expressed in breast cancers. Authors could include some information on these targets and the review will be gaining a lot of interest from the readers.
Author Response
Responses to the reviewers
Reviewer 1
The authors have written a well summarized review on precision and targeted therapeutics for breast cancer subtypes including the TNBC.
Ans. Thank you.
The review covers good information for Her2 based targeted therapeutics and its current challenges. Also including the various immune check point inhibitors such as the PI3K/AKT/mTOR. The review also mentions the PD-L1 and EGFR targeting.
However the authors could also include some literature for CD44 receptor and how this could be used for precision based medicines for TNBC and breast cancers include the breast cancer stem cells. CD44 receptor has been widely used for targeted therapeutics in TNBC using nanomedicine (https://doi.org/10.1021/acs.molpharmaceut.2c00439).
Ans. Included in the revised version.
Similarly nanomedicines particularly the cubosome nanocarriers have been used for precision based therapy against the carcinoembryonic based antigen (CEA). CEA is also expressed in breast cancers. Authors could include some information on these targets and the review will be gaining a lot of interest from the readers.
Ans: Included in the revision.

Reviewer 2 Report
The paper by Md Abdus Subhan et al comprises a review addressing precision medicine treatment for metastatic breast cancer and triple-negative breast cancer. It is an actual and interesting topic that can attract researchers and doctors in the field. However, there are several observations that need to be addressed. Of note, the lack of line numbers in the manuscript made it difficult to describe the observations. Therefore, I referred to pages and sections.
1) Title.
The title is not really accurate for the theme explored in the Review. The title is: “Recent advances with precision medicine treatment for metastatic breast cancer and triple-negative breast cancer”. However, all of section 2 is devoted to HER2+ breast cancer, whether it is metastatic or not. On the other hand, precision medicine for patients with ER+, PR+ metastatic breast cancer was barely reviewed. I suggest changing the title to adequately reflect the content of the review.
2) General overview and Section 1.
The sections of the review are somehow not fluently integrated, giving the impression that different persons wrote each one, without communicating between them. Some sections are well drafted, while others are not so much. Also, while reading, it seems that pieces of the review have been put together disregarding what has already been discussed or abbreviated before. In some instances, the acronyms are used without previous description, while in others, they are described in multiple parts of the text, in a repetitive manner. Authors should address these flaws.
For example, in section 1 (Introduction), authors described very well what is precision medicine, as well as which are the different types of breast cancer, with a focus on druggable targets present in each subtype. Nevertheless, at the beginning of sections 2 and 3, precision medicine and the breast cancer subtypes are described again. Please try to integrate the review considering what has been previously discussed, to avoid repetitions.
· The acronym HER2 is used widely through section 1, but it is only defined in section 2. Similarly, other abbreviations and acronyms need to be revised. All acronyms’ definitions must be done at first mention.
· Figure 1, ESCAT ranking: Is there no Tier Ib? You can check in: https://doi.org/10.1007/s12254-022-00800-1
· Please define PT and ICI at first mention.
· In the phrase: “Lapatinib binds reversibly to the ATP-binding site of both receptor and inhibits receptor phosphorylation”, add an “s” to receptor, for plural.
· Please define HDAC and HDACi at first mention. And do not repeat this definition after. Of note, HDACi is only defined several pages after, on page 18. Please solve this inconsistency.
· Define PARPi.
· Please explain what is “synthetic lethality”, to help the reader, and how does PARPi induce this phenomenon.
· Please explain more in-depth how PARPi is related to “Trapp in DNA”
3) Section 2.
· All section 2.2 needs revision and rephrasing, the paragraphs are either truncated or drafted in a confusing manner. Please revise this section carefully. For instance, in the following phrases, the wording is unclear, making it difficult to understand:
“The HER2 mAbs have been designed that binds to the HER2 receptor with higher specificity than trastuzumab, and have the ability to binds with additional epitopes to improve activity or elicit a greater immunological response, include trastuzumab, mar-getuximab, ZW25, and PRS-343 »
«The PRS-343 showed the clustering of CD137 by combining CD137 positive T cells to HER2-positive cancer cells leads to enhanced stimulation of tumor antigen-specific T cells»
«Both trials have allowed for patients with advanced solid tumors»
· In section 2.3, it is wrong to say that «Trastuzumab deruxtecan is a topoisomerase I inhibitor». Actually, trastuzumab-deruxtecan contains two different components, the monoclonal antibody trastuzumab that binds HER2, and the anticancer drug deruxtecan, which helps kill cancer cells. It is this last component that acts as a topoisomerase inhibitor. In the review, this is not clearly stated. It would be interesting for the reader if a more in-depth analysis was performed regarding the ADCs, and not only mention which combinations are being tested. The way it is written is dull and does not provide much insight. A short and punctual description of the mechanisms of action of the ADCs could make this section more interesting and less descriptive.
· The phrase: “ZW49 demonstrated good in-vitro studies” does not make any sense. Please rephrase.
· Please give some context/explanation to the phrase: “The in-vivo studies also demonstrated high and low HER2 levels in patient-derived xenograft models”.
4) Section 3.
· Please rephrase: “This information is then used to guide the guide treatment for each individual patient”.
· Define MBC.
5) Section 4.
This section is interesting and well-written.
In the title of this section, change: “of TNBC” to “for TNBC”.
Do not describe again what “TNBC” and “HER2” acronyms stand for. These terms have already been defined.
6) Section 5.
· The mentioned RTKs´ acronyms like VEGFR, IGFR, etc, have not been defined. Please define them all.
· Please add the reference for the phrase: “A dual EGFR/HER2 RTKi, effective in HER2+ breast cancer, however was ineffective in TNBC.”
· The phrase: “PI3K inhibitors may decrease BRCA1 and BRCA2” is incomplete.
· Please consider that, in the context of pathological cyclin D1-associated kinase activity, as in cancer, CDK4/CDK6 expression can be deranged, overexpressed or altered. In this section of the review, authors only mention that: « CDK4, CDK6 expressions are common in TNBC”. However, the expression of these CDKs is also observed in normal tissue.
· The title of this subsection is: “Potential applications of precision medicine therapy for TNBC”. However, treatment for hormone receptor-positive and HER2+ breast cancer patients was also addressed. I suggest restricting the information to TNBC, as the title depicts, otherwise, it would need more information and would be too long.
· It is not known what “ICIs” refers to. Please define.
· This section needs drafting correction. For instance, the phrase: “PARP inhibitor including anti-VEGFR included" needs rewriting.
· Please correct the misspelling error in the phrase: “PARPi in combinatio1n with PD-L1 inhibitors could sensitize the PARPi against TNBC cells.”
· The Paragraph: “Likewise, AKT inhibitor drug, AZD5363 in patients with AKT1 E17K-mutant ER+ breast cancer and MK-2206 in patients with PIK3CA-mutant ER+ and HER2– breast cancer was investigated in neo-adjuvant settings for individualized treatment options [182-184]”, does not tell the reader the outcome of the studies. Introducing a phrase with the achieved result/conclusions in these studies could make this section more appealing.
Author Response
Reviewer 2
The paper by Md Abdus Subhan et al comprises a review addressing precision medicine treatment for metastatic breast cancer and triple-negative breast cancer. It is an actual and interesting topic that can attract researchers and doctors in the field. However, there are several observations that need to be addressed. Of note, the lack of line numbers in the manuscript made it difficult to describe the observations. Therefore, I referred to pages and sections.
Ans: Thank you.
1) Title.
The title is not really accurate for the theme explored in the Review. The title is: “Recent advances with precision medicine treatment for metastatic breast cancer and triple-negative breast cancer”. However, all of section 2 is devoted to HER2+ breast cancer, whether it is metastatic or not. On the other hand, precision medicine for patients with ER+, PR+ metastatic breast cancer was barely reviewed. I suggest changing the title to adequately reflect the content of the review.
Ans: Title has been modified.
2) General overview and Section 1.
The sections of the review are somehow not fluently integrated, giving the impression that different persons wrote each one, without communicating between them. Some sections are well drafted, while others are not so much. Also, while reading, it seems that pieces of the review have been put together disregarding what has already been discussed or abbreviated before. In some instances, the acronyms are used without previous description, while in others, they are described in multiple parts of the text, in a repetitive manner. Authors should address these flaws.
For example, in section 1 (Introduction), authors described very well what is precision medicine, as well as which are the different types of breast cancer, with a focus on druggable targets present in each subtype. Nevertheless, at the beginning of sections 2 and 3, precision medicine and the breast cancer subtypes are described again. Please try to integrate the review considering what has been previously discussed, to avoid repetitions.
- The acronym HER2 is used widely through section 1, but it is only defined in section 2. Similarly, other abbreviations and acronyms need to be revised. All acronyms’ definitions must be done at first mention.
Ans. Modified.
- Figure 1, ESCAT ranking: Is there no Tier Ib? You can check in: https://doi.org/10.1007/s12254-022-00800-1
Ans. Reference was added and mentioned un the text.
- Please define PT and ICI at first mention.
Ans. Defined.
- In the phrase: “Lapatinib binds reversibly to the ATP-binding site of both receptor and inhibits receptor phosphorylation”, add an “s” to receptor, for plural.
Ans. Performed.
- Please define HDAC and HDACi at first mention. And do not repeat this definition after. Of note, HDACi is only defined several pages after, on page 18. Please solve this inconsistency.
Ans. Addressed.
- Define PARPi.
- Please explain what is “synthetic lethality”, to help the reader, and how does PARPi induce this phenomenon.
- Please explain more in-depth how PARPi is related to “Trapp in DNA”
Ans. Addressed. Described in the text.
3) Section 2.
- All section 2.2 needs revision and rephrasing, the paragraphs are either truncated or drafted in a confusing manner. Please revise this section carefully. For instance, in the following phrases, the wording is unclear, making it difficult to understand:
“The HER2 mAbs have been designed that binds to the HER2 receptor with higher specificity than trastuzumab, and have the ability to binds with additional epitopes to improve activity or elicit a greater immunological response, include trastuzumab, mar-getuximab, ZW25, and PRS-343 »
«The PRS-343 showed the clustering of CD137 by combining CD137 positive T cells to HER2-positive cancer cells leads to enhanced stimulation of tumor antigen-specific T cells»
«Both trials have allowed for patients with advanced solid tumors»
- In section 2.3, it is wrong to say that «Trastuzumab deruxtecan is a topoisomerase I inhibitor». Actually, trastuzumab-deruxtecan contains two different components, the monoclonal antibody trastuzumab that binds HER2, and the anticancer drug deruxtecan, which helps kill cancer cells. It is this last component that acts as a topoisomerase inhibitor. In the review, this is not clearly stated. It would be interesting for the reader if a more in-depth analysis was performed regarding the ADCs, and not only mention which combinations are being tested. The way it is written is dull and does not provide much insight. A short and punctual description of the mechanisms of action of the ADCs could make this section more interesting and less descriptive.
- The phrase: “ZW49 demonstrated good in-vitro studies” does not make any sense. Please rephrase.
- Please give some context/explanation to the phrase: “The in-vivo studies also demonstrated high and low HER2 levels in patient-derived xenograft models”.
Ans. Section 2 has been modified according to your suggestion.
4) Section 3.
- Please rephrase: “This information is then used to guide the guide treatment for each individual patient”.
- Define MBC.
Ans. Performed.
5) Section 4.
This section is interesting and well-written.
In the title of this section, change: “of TNBC” to “for TNBC”.
Ans. Modified.
Do not describe again what “TNBC” and “HER2” acronyms stand for. These terms have already been defined.
6) Section 5.
- The mentioned RTKs´ acronyms like VEGFR, IGFR, etc, have not been defined. Please define them all.
Ans. Added.
- Please add the reference for the phrase: “A dual EGFR/HER2 RTKi, effective in HER2+ breast cancer, however was ineffective in TNBC.”
Ans. Reference added.
- The phrase: “PI3K inhibitors may decrease BRCA1 and BRCA2” is incomplete.
Ans. Modified.
- Please consider that, in the context of pathological cyclin D1-associated kinase activity, as in cancer, CDK4/CDK6 expression can be deranged, overexpressed or altered. In this section of the review, authors only mention that: « CDK4, CDK6 expressions are common in TNBC”. However, the expression of these CDKs is also observed in normal tissue.
Ans. Addressed this issue.
- The title of this subsection is: “Potential applications of precision medicine therapy for TNBC”. However, treatment for hormone receptor-positive and HER2+ breast cancer patients was also addressed. I suggest restricting the information to TNBC, as the title depicts, otherwise, it would need more information and would be too long.
Ans. Removed and modified.
- It is not known what “ICIs” refers to. Please define.
Ans. Addressed.
- This section needs drafting correction. For instance, the phrase: “PARP inhibitor including anti-VEGFR included" needs rewriting.
Ans. Modified.
- Please correct the misspelling error in the phrase: “PARPi in combinatio1n with PD-L1 inhibitors could sensitize the PARPi against TNBC cells.”
Ans. Corrected.
Section 6
- The Paragraph: “Likewise, AKT inhibitor drug, AZD5363 in patients with AKT1 E17K-mutant ER+ breast cancer and MK-2206 in patients with PIK3CA-mutant ER+ and HER2– breast cancer was investigated in neo-adjuvant settings for individualized treatment options [182-184]”, does not tell the reader the outcome of the studies. Introducing a phrase with the achieved result/conclusions in these studies could make this section more appealing.
Ans. Addressed. Modified.

Round 2
Reviewer 2 Report
The authors made a good effort in redrafting their manuscript, and the observations were correctly addressed. I believe it is now suitable for publication. Only one minor suggestion:
I feel the title is a little bit repetitive as it is, I suggest the following: “Recent advances with precision medicine treatment for breast cancer”
Author Response
Dear reviewer
Thank you for your inspiring comment on our manuscript.
Sincerely yours
Md Abdus Subhan